# Relationship between Cerebrospinal Fluid Matrix Metalloproteinases Levels and Brain Amyloid Deposition in Mild Cognitive Impairment

**DOI:** 10.3390/biom11101496

**Published:** 2021-10-11

**Authors:** Yuuki Sasaki, Noriyuki Kimura, Yasuhiro Aso, Kenichi Yabuuchi, Miki Aikawa, Etsuro Matsubara

**Affiliations:** 1Department of Neurology, Faculty of Medicine, Oita University, Oita 879-5593, Japan; sasaki-y@oita-u.ac.jp (Y.S.); yasuhiroaso@oita-u.ac.jp (Y.A.); kyabuuchi@oita-u.ac.jp (K.Y.); etsuro@oita-u.ac.jp (E.M.); 2Kameda Medical Center, Chiba 296-8602, Japan; am12ki1i@gmail.com

**Keywords:** cerebrospinal fluid, ^11^C-Pittsburgh compound B positron emission tomography, ^18^F-fluorodeoxyglucose positron emission tomography, matrix metalloproteinases, mild cognitive impairment, tissue inhibitor of metalloproteinases, white matter lesions

## Abstract

This study aimed to explore whether cerebrospinal fluid (CSF) levels of matrix metalloproteinases (MMPs), and their inhibitors (TIMPs) were associated with brain amyloid deposition, cortical glucose metabolism, and white matter lesions (WMLs) in individuals with amnestic mild cognitive impairment (MCI). A total of 33 individuals with amnestic MCI (mean age, 75.6 years) underwent ^11^C-Pittsburgh compound B positron emission tomography (PiB-PET), ^18^F-fluorodeoxyglucose positron emission tomography, magnetic resonance imaging or computed tomography, and CSF analysis. PET uptake of the frontal and temporoparietal lobes and posterior cingulate gyrus was assessed using the cerebellar cortex as the reference region. WMLs were assessed by the Fazekas scale. CSF levels of MMPs and TIMPs were measured with bead-based multiplex assays. After adjusting for covariates, multiple linear regression analysis showed that CSF levels of MMP-2 were negatively correlated with global PiB uptake (*p* = 0.035), especially in the parietotemporal lobe and posterior cingulate gyrus (*p* = 0.016 and *p* = 0.041, respectively). Moreover, CSF levels of MMP-7 were positively correlated with the severity of WMLs (*p* = 0.033). CSF levels of MMP-2 and MMP-7 are associated with brain amyloid deposition and severity of WMLs, respectively. These findings provide valuable insights into the role of MMPs in amyloid β catabolism and blood–brain barrier integration at the MCI stage.

## 1. Introduction

Matrix metalloproteinases (MMPs) are a family of zinc- and calcium-dependent endopeptidases secreted primarily from neurons and glia [1]. Their activation or activity is regulated by members of the tissue inhibitor of the metalloproteinase (TIMP) family [2]. MMPs cleave most extracellular matrix proteins and are implicated in various physiological functions, such as tissue remodeling, inflammation, and angiogenesis [1,3]. Recent evidence suggests that MMPs and TIMPs contribute to the pathogenesis of Alzheimer’s disease (AD) [4,5,6,7,8]. The two-hit vascular hypothesis of AD proposes that aging and vascular risk factors lead to the dysregulation of the blood–brain barrier (BBB) and oligemia, resulting in impaired amyloid β (Aβ) clearance and increased Aβ production [9,10]. MMPs is highly expressed in the postmortem AD brain tissue [5]. Moreover, MMPs degrade Aβ protein [4,5], tight junction, and basement membrane proteins [11,12,13,14]. However, it remains unclear whether MMPs have protective or detrimental roles in the pathogenesis of AD. Previously, we reported that plasma levels of MMPs and TIMPs were not associated with brain amyloid deposition but the severity of white matter lesions (WMLs) in individuals with amnestic mild cognitive impairment (MCI) [15]. However, cerebrospinal fluid (CSF) more precisely reflects biochemical changes in the brain compared with blood [16]. In fact, several studies showed that the CSF levels of MMPs and TIMPs were correlated with AD biomarkers [17,18,19,20]. To our knowledge, few studies have investigated the relationship between CSF levels of MMPs and TIMPs and imaging biomarkers, such as ^11^C-Pittsburgh compound B positron emission tomography (PiB-PET), ^18^F-fluorodeoxyglucose positron emission tomography (FDG-PET), and magnetic resonance imaging (MRI) or computed tomography (CT). Structural and functional imaging techniques are useful for identifying individuals at increased risk of developing AD. In particular, PiB-PET can detect brain amyloid deposition and predict conversion from MCI to AD [21,22]. WMLs on MRI or CT are frequently observed in patients with AD and influence cognitive function or cerebral perfusion [23,24]. BBB disruption and chronic hypoperfusion are the main causes of WMLs [25,26]. We hypothesized that CSF levels of MMPs and TIMPs may be associated with brain amyloid deposition and BBB disruption, resulting in white matter degeneration at the MCI stage. Therefore, this study aimed to examine the potential associations among CSF levels of MMPs and TIMPs, brain amyloid deposition, cortical glucose metabolism, and WMLs in individuals with amnestic MCI.

## 2. Materials and Methods

### 2.1. Subjects

Thirty-three individuals with amnestic MCI (11 men and 22 women; mean age, 75.6 ± 5.4 years) were recruited from outpatients at the Department of Neurology, Oita University Hospital, between 2012 and 2018. The diagnosis of amnestic MCI was made according to previous study [15]. All individuals had an ischemic score of 4 or lower on Hachinski’s scale [27]. Evaluation of cognitive function, brain MRI or CT, PiB- and FDG-PET, and CSF measurements of MMPs and TIMPs were performed in all individuals. Information regarding age, sex, and education level was collected. Mini-Mental State Examination (MMSE) was used to evaluate cognitive function. Assessment of vascular risk factor data, including hypertension, diabetes mellitus, and hypercholesterolemia, was based on detailed clinical history and medication use.

### 2.2. Multiplex Assay

Lumbar puncture was performed between 9:00 and 10:00 a.m. with fasted patients in the lateral recumbent position. CSF samples were collected after centrifugation at 1500× *g* for 10 min. The supernatant was frozen and stored in 1 mL aliquots at −80 °C until use. We performed multiplex bead-based assays to determine the CSF levels of MMPs and TIMPs. CSF concentrations of nine MMPs and two TIMPs were simultaneously measured using the Bio-Plex Pro Human MMP panel, including MMP-1, MMP-2, MMP-3, MMP-7, MMP-8, MMP-9, MMP-10, MMP-12, and MMP-13 (171-AM001M; Bio-Rad) and Milliplex Map Human TIMP Magnetic Bead Panel 1, including TIMP-1 and TIMP-2 (HTMP1MAG-54K; Millipore). Data analysis was conducted on Bio-Plex 200 suspension array system (Bio-Rad, Hercules, CA, USA) with Bio-Plex Manager Software version 6.0. All CSF samples were run in duplicate on the same plate. These results were represented as the mean fluorescence intensity.

### 2.3. White Matter Lesion Assessment

WMLs were evaluated by MRI for 30 individuals and by CT for 3 individuals. Axial T1-weighted and T2-weighted MR images were obtained using a 3.0 T Siemens MRI scanner (Magnetom Verio; Siemens, Erlangen, Germany). The imaging parameters were as follows: repetition time (TR) of 1900 ms/echo time (TE) of 2.53 ms for T1-weighted images and TR 4000–5000 ms/TE 80–100 ms for T2-weighted images. Non-enhanced brain CT was performed with 3 mm continuous slices using Biograph 40 (Siemens, Erlangen, Germany). WML severity was assessed on T2-weighted images or CT scans using the Fazekas scale [28], according to previous studies [15,24]. Fazekas scale is widely used to classify periventricular or deep WMLs as graded 0 (absent) through 3 (severe). Briefly, the severity of periventricular WMLs was graded as follows: 0 for absent, 1 for “caps” or pencil-thin lining, 2 for smooth “halo”, and 3 for irregular periventricular lesions extending into the deep white matter. The severity of deep WMLs was graded as follows: 0 for absent, 1 for punctate foci, 2 for the beginning of confluent foci, and 3 for large confluent areas. In this study, the Fazekas scale was determined as the sum of the periventricular and deep WMLs scores (Figure 1). All images were independently evaluated by two neurologists blinded to the medical information. They discussed together for consensus in cases of disagreement. The severity of WMLs in all individuals was classified as grade 0, 1, or 2 on the Fazekas scale. Therefore, individuals with severe WMLs that might indicate vascular dementia were not included in this study.

### 2.4. Apolipoprotein E Phenotype

Human Apolipoprotein E4/Pan-APOE ELISA kit (MBL Co., Ltd., Woburn, MA, USA) was used to determine apolipoprotein E (*APOE*) phenotype. The amount of *APOE4* and total *APOE* was measured with high sensitivity using an affinity-purified polyclonal antibody against *APOE* and a monoclonal antibody against *APOE4* by sandwich enzyme-linked immunosorbent assay. The ratio of *APOE* and *APOE4* levels can discriminate between homozygotes (ε4/ε4) or heterozygotes (ε2/ε4, ε3/ε4) of *APOE4* phenotypes and non-*APOE4* zygotes (ε2/ε2, ε3/ε3, and ε2/ε3) [29,30].

### 2.5. PET

Static PiB-PET and FDG-PET studies were conducted in three-dimensional scanning mode using a Siemens Biograph mCT PET scanner (Siemens) as previously described [15,30]. The reagents were provided by the PET center of our hospital. Participants received an intravenous bolus injection of PiB [mean ± standard deviation (SD) of radioactivity = 523 ± 47 MBq] and subsequent saline flush in the PiB-PET study. Then, PET images were acquired 50–70 min after the injection. To perform FDG-PET, all participants kept close their eyes and relaxed in a dimly lit room for 10 min before the injection. Then, they received a bolus intravenous injection of 3.0 MBq/kg FDG and subsequent saline flush. PET images were acquired 40–60 min after the injection. The radiation in pre-pose and post-dose samples was measured using a radiation detector for the calculation of injected dose in each participant. All imaging data were reconstructed into a 3.0-mm thick slice, on a 256 × 256 matrix, and at 3.0× magnification with an ordered-subset expectation maximization that includes four iterations and 12 subsets. The reconstructed images had a pixel size of 1.06 mm. The PiB-PET and FDG-PET scans were both spatially normalized to the Montreal Neurological Institute reference space through a customized PET template using Statistical Parametric Mapping version 8 (Wellcome Trust Centre for Neuroimaging, https://www.fil.ion.ucl.ac.uk/spm/) (accessed date 7 October 2021). Regions of interest (ROIs), such as the frontal lobe, temporoparietal lobe, posterior cingulate gyrus, and cerebellum, were determined using the MarsBaR (MRC Cognition and Brain Sciences Unit) ROI toolbox for Statistical Parametric Mapping as described previously [24]. These ROIs were established as areas with amyloid deposition and decreased cortical glucose metabolism in patients with AD [31,32]. PiB and FDG uptake was assessed by a standardized uptake value ratio (SUVR). The ROI values were averaged across both hemispheres. The globally standardized uptake value ratio of FDG-PET and PiB-PET was represented as a single mean value for all regions combined (Figure 2 and Figure 3). Higher amyloid uptake was determined using a mean cortical SUVR of 1.4 or higher as the cut off [30].

### 2.6. Statistical Analysis

We performed a multiple regression model to investigate the relationship among the CSF levels of MMPs and TIMPs, the PiB and FDG uptake values, and Fazekas scale scores after adjusting for covariates, such as age, sex, education level, frequency of each vascular risk factor, and APOE4 status. Additionally, the relationship between the CSF levels of MMPs and TIMPs and PiB uptake values in each ROI was assessed by a multiple regression model. Statistical analyses were performed using IBM SPSS Statistics for Windows version 25.0 (IBM Corp., Armonk, NY, USA). A *p*-value < 0.05 was considered significant.

## 3. Results

### 3.1. Clinical and Demographic Characteristics

Table 1 summarizes the sociodemographic factors, PET imaging, and Fazekas scale of individuals with amnestic MCI. The mean participant age was 75.6 ± 5.4 years; 11 (33.3%) were males, and 22 (66.7%) were females. The mean education level of 11.4 ± 1.9 years. The mean MMSE score was 24.8 ± 2.0. Additionally, 11 individuals (33.3%) were APOE4 carriers. The frequencies of hypertension, diabetes mellitus, and hypercholesterolemia were 60.6%, 9.1%, and 51.5%, respectively. 17 individuals (51.5%) were included in the higher PiB subgroup based on a PiB-PET SUVR cutoff of 1.4, while 20 individuals (60.6%) showed WMLs on MRI or CT. The severity of WMLs in all individuals was classified as grade 0, 1, or 2 on the Fazekas scale. The severity of periventricular WMLs was grade 1 in 9 individuals (27.3%) and grade 2 in 7 individuals (21.2%); the scores for deep WMLs were grade 1 in 7 individuals (21.2%) and grade 2 in 7 individuals (21.2%). There were no significant differences in CSF levels of MMPs and TIMPs between individuals with and without *APOE4* or between individuals with and without vascular risk factors (Appendix A).

### 3.2. Relationship among CSF Levels of MMPs and TIMPs and PiB and FDG Uptake Values and Fazekas Scale Scores

The results of the multiple regression models among CSF levels of MMPs and TIMPs, PET imaging variables, and Fazekas scale scores are summarized in Table 2 and Table 3. MMP-1, MMP-3, MMP-8, MMP-9, MMP-10, and MMP-13 were below the limit of detection in most samples. The remaining three MMPs (MMP-2, MMP-7, and MMP-12) and two TIMPs (TIMP-1 and TIMP-2) were detectable and included in the analysis. CSF levels of MMP-2 were negatively correlated with global PiB uptake after adjusting for covariates [*β* = −0.414; 95% confidence interval (CI), −0.796 to −0.032, *p* = 0.035; Table 2 and Figure 4A). Particularly, CSF levels of MMP-2 were significantly correlated with PiB uptake in the posterior cingulate gyrus and parietotemporal lobe (*β* = −0.399; 95% CI, −0.782 to −0.017, *p* = 0.041 and *β* = −0.476; 95% CI, −0.856 to −0.096, *p* = 0.016, respectively; Table 3 and Figure 5A,B). Moreover, the CSF levels of MMP-7 were positively correlated with Fazekas scale score (*β* = 0.419; 95% CI, 0.036 to 0.802, *p* = 0.033; Table 2, Figure 4B). No significant correlation was found between the CSF levels of MMPs and TIMPs and global and regional FDG uptake.

## 4. Discussion

To the best of our knowledge, this study is the first to investigate whether CSF levels of MMPs and TIMPs are associated with brain amyloid deposition, glucose metabolism, and WML severity in individuals with amnestic MCI. Out of 33 subjects, 17 (51.5%) had higher PiB uptake and 20 (60.6%) had WMLs. The frequency of higher amyloid deposition is reported to be 54.6% in individuals with MCI aged 75–80 years [33] and that of WMLs to be 5–87% in individuals aged 65 years and older, depending on the clinical characteristics and the method used to assess WMLs [34]. Moreover, two prior studies reported that WMLs were observed in approximately 70–80% of patients with AD [35,36]. Therefore, the frequency of individuals with higher PiB uptake or WMLs in the present study is consistent with findings from previous research. We discovered that CSF levels of MMP-2 were negatively correlated with PiB uptake, especially in the posterior cingulate gyrus and parietotemporal lobe. Moreover, CSF levels of MMP-7 were positively correlated with the Fazekas scale score. These results provide new and valuable insights into the role of MMPs in brain amyloid deposition and white matter degeneration at the MCI stage.

The most interesting finding of our study was the negative correlation between MMP-2 CSF levels and brain amyloid deposition. This finding supports the protective role of MMP-2 in the accumulation of Aβ in the brain [14]. MMPs are classified into gelatinases (MMP-2 and MMP-9), matrilysin (MMP-7), stromelysins (MMP-3 and MMP-10), collagenases (MMP-1, MMP-8, and MMP-13), membrane-type MMPs, and other MMPs based on structure and substrate specificity [1,3,16,37]. Several studies have reported an association between CSF levels of MMPs and AD pathology. MMP-2, MMP-3, and MMP-9 are implicated in the degradation of Aβ, and their expression is increased near amyloid plaques [38,39,40]. Other studies have found that MMP-2 levels are lower in patients with AD compared to controls [8,41]. Moreover, it was reported that decreased levels of MMP-2 and MMP-3 are associated with low Aβ levels [41], and increased levels of MMP-3, MMP-9, and MMP-10 are associated with high total tau or phosphorylated tau levels in CSF [18,32,42]. We showed an association between decreased MMP-2 CSF levels and increased PiB uptake, which is consistent with previous studies. These findings suggest that decreased MMP-2 levels in CSF reflect impaired Aβ degradation, thereby leading to senile plaque formation in the MCI stage.

Another interesting finding of this study was the positive correlation between MMP-7 CSF levels and Fazekas scale score in our cohort. MMP-7 is secreted from microglia [43]. MMP-2, MMP-7, and MMP-9 are implicated in BBB disruption through the degradation of tight junction proteins and basement membrane components, such as laminin, entactin, and type IV collagen [4,14,18,44,45]. Moreover, MMP-7 CSF level is associated with major depressive disorder, multiple sclerosis, and human immunodeficiency virus dementia [43,46,47]. We showed an association between MMP-7 in CSF and WMLs in individuals with amnestic MCI for the first time. These findings suggest that MMP-7 is involved in BBB disruption, which leads to white matter degeneration at the MCI stage. Although a previous in vitro study reported that MMP-7 may degrade Aβ1-42 to result in the prevention of Aβ aggregation [48], our results showed no correlation between MMP-7 and brain amyloid deposition. One possible explanation for this discrepancy is that PiB-PET mainly detects insoluble fibrillar Aβ deposits [49].

Effective disease-modifying drugs are urgently needed to prevent the onset or slow the progression of AD. Potential targets for novel AD drugs include amyloid, tau, inflammation, metabolism, and neuroprotection [50]. Moreover, upregulation of lysosomal hydrolase and cathepsin B has been identified as a potential disease-modifying therapy in transgenic mouse models of AD [51,52]. Our results suggest that MMPs are a potential therapeutic target for slowing the progression of MCI to AD. However, activation of MMPs appears to have both a beneficial role in Aβ catabolism and a detrimental role in BBB integration [53]. Therefore, we emphasize that caution must be taken when using MMPs as a therapeutic target due to their complex role in the pathology of AD.

In this study, several MMPs, namely, MMP-1, MMP-3, MMP-8, MMP-9, MMP-10, and MMP-13, were below the limit of detection in CSF. A number of studies have reported variable findings regarding the CSF levels of MMPs and TIMPs in patients with AD, such as increased or decreased levels of MMP-3 [8,32,41,42], increased or undetectable levels of MMP-9 and MMP-10 [42,54], increased or decreased levels of TIMP-1 [32,42], and increased levels of TIMP-2 [55] compared to controls. These confounding findings may be due to differences in subject selection and the measurement methods used to determine MMP and TIMP.

There are several limitations to this study. Our study cohort was diagnosed based on clinical findings. Therefore, our recruited participants may have concomitant suspected non-Alzheimer’s pathology. It is not possible to determine the causal direction of the association among MMPs, brain amyloid deposition, and WMLs. Moreover, the number of individuals recruited in our study was modest. Therefore, future longitudinal studies with larger sample sizes are required to evaluate the temporal change in CSF levels of MMPs with the progression of brain amyloid deposition and white matter degeneration in AD.

## 5. Conclusions

We found that CSF levels of MMP-2 are associated with brain amyloid deposition, and MMP-7 CSF levels are associated WMLs. The findings of the present study suggest that MMPs play an important role in amyloid catabolism and BBB disruption at the MCI stage.

## Figures and Tables

**Figure 1 biomolecules-11-01496-f001:**
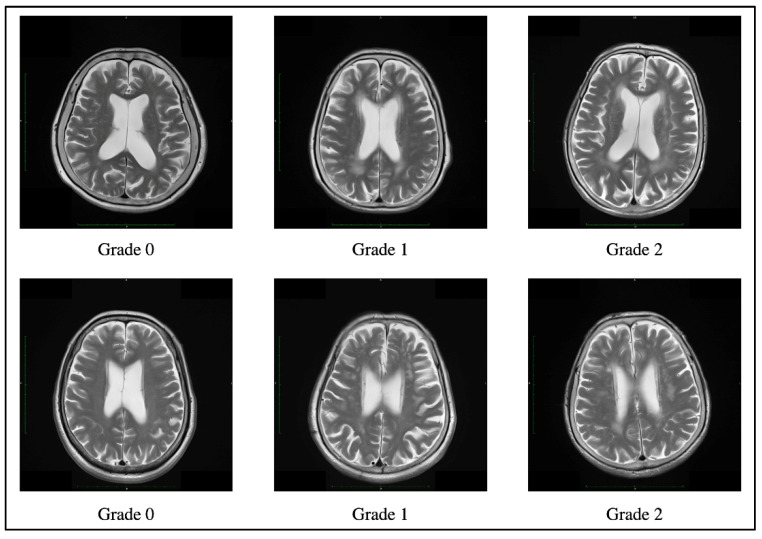
Example of a Fazekas scale score of 0, 1, and 2 periventricular hyperintensity (**top**) and deep white matter hyperintensity (**bottom**) on T2-weighted images.

**Figure 2 biomolecules-11-01496-f002:**
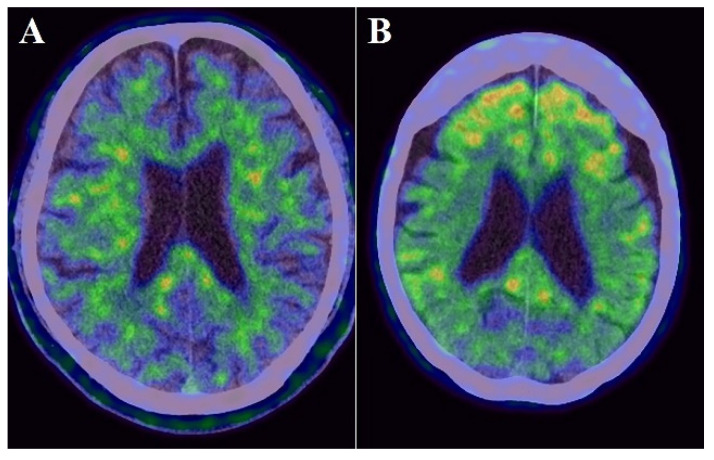
Examples images of an individual with lower PiB uptake ((**A**): cortical SUVR value = 0.804) and with higher PiB uptake ((**B**): cortical SUVR value = 2.584).

**Figure 3 biomolecules-11-01496-f003:**
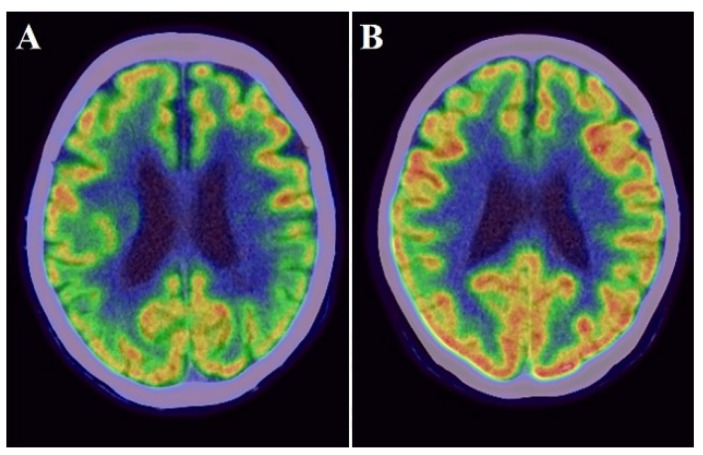
Examples images of an individual with lower FDG uptake ((**A**): cortical SUVR value = 0.709) and with higher FDG uptake ((**B**): cortical SUVR value = 0.874).

**Figure 4 biomolecules-11-01496-f004:**
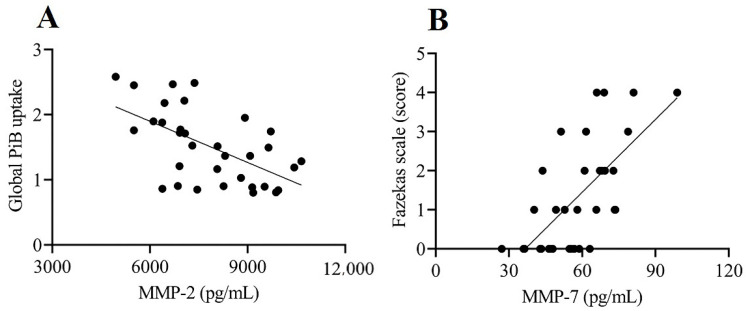
Relationship between CSF levels of MMPs and PiB uptake and Fazekas scale score. (**A**) MMP-2 CSF levels are negatively correlated with global PiB uptake (*β* = −0.414; 95% confidence interval, −0.796 to −0.032, *p* = 0.035). (**B**) MMP-7 CSF levels are positively correlated with Fazekas scale score (*β* = 0.419; 95% confidence interval, 0.036 to 0.802, *p* = 0.033).

**Figure 5 biomolecules-11-01496-f005:**
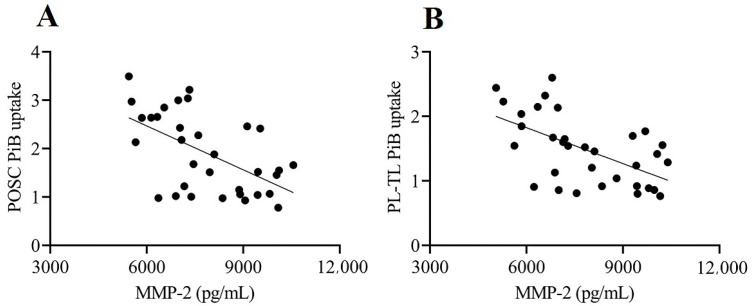
Relationship between CSF levels of MMP-2 and PiB uptake in the brain ROI. CSF levels of MMP-2 are negatively correlated with PiB uptake in the (**A**) posterior cingulate gyrus (*β* = −0.399; 95% CI, −0.782 to −0.017, *p* = 0.041) and (**B**) parietotemporal lobe (*β* = −0.476; 95% CI, −0.856 to −0.096, *p* = 0.016). PL-TL—parietotemporal lobe; POSC—posterior cingulate gyrus.

**Table 1 biomolecules-11-01496-t001:** Clinical and demographic characteristics.

Characteristic	Cohort (N = 33)
Age, mean (SD), years	75.6 (5.4)
Sex (M/F)	11:22
Education level, mean (SD), years	11.4 (1.9)
APOE4, no. (%)	11 (33.3%)
Hypertension (%)	20 (60.6%)
Diabetes (%)	3 (9.1%)
Hypercholesterolemia (%)	17 (51.5%)
MMSE, mean (SD), score	24.8 (2.0)
FL PiB uptake, mean (SD)	1.46 (0.55)
POSC PiB uptake, mean (SD)	1.91 (0.81)
PL-TL PiB uptake, mean (SD)	1.48 (0.53)
Global PiB uptake, mean (SD)	1.51 (0.56)
Global FDG uptake, mean (SD)	0.88 (0.08)
Fazekas scale, mean (SD)	1.33 (1.41)

APOE—apolipoprotein E; F—female; FDG—18^F^-fluorodeoxyglucose; FL—frontal lobe; M—male; MMSE—Mini-Mental State Examination; PiB—^11^C-Pittsburgh Compound B; PL-TL—parietotemporal lobe; POSC—posterior cingulate gyrus; SD—standard deviation.

**Table 2 biomolecules-11-01496-t002:** Multiple regression model between CSF MMPs and TIMPs levels and global PiB uptake, global FDG uptake, and Fazekas scale score.

CSF Level	Global PiB Uptake		Global FDG Uptake		Fazekas Scale Score	
	β (95% CI)	*p*	β (95% CI)	*p*	β (95% CI)	*p*
MMP-2	−0.414 (−0.796, −0.032)	0.035 ^1^	0.174 (−0.247, 0.595)	0.403	0.106 (−0.304, 0.517)	0.598
MMP-7	−0.259 (−0.674, 0.156)	0.209	0.105 (−0.329, 0.539)	0.623	0.419 (0.036, 0.802)	0.033 ^1^
MMP-12	−0.139 (−0.531, 0.253)	0.473	−0.069 (−0.471, 0.333)	0.724	−0.04 (−0.43, 0.349)	0.832
TIMP-1	−0.177 (−0.587, 0.233)	0.382	−0.126 (−0.547, 0.295)	0.544	0.253 (−0.143, 0.649)	0.20
TIMP-2	−0.103 (−0.519, 0.313)	0.615	−0.14 (−0.561, 0.282)	0.50	0.166 (−0.239, 0.571)	0.406

CI—confidence interval; CSF—cerebrospinal fluid; MMP—matrix metalloproteinase; TIMP—tissue inhibitor of metalloproteinase. ^1^ A *p*-value < 0.05 was considered statistically significant.

**Table 3 biomolecules-11-01496-t003:** Multiple regression model between CSF MMPs and TIMPs levels and PiB uptake in each brain ROI.

CSF Level	FL		POSC		PL-TL	
	β (95% CI)	*p*	β (95% CI)	*p*	β (95% CI)	*p*
MMP-2	−0.376 (−0.76, 0.008)	0.055	−0.399 (−0.782, −0.017)	0.041 ^1^	−0.476 (−0.856, −0.096)	0.016 ^1^
MMP-7	−0.226 (−0.64, 0.187)	0.27	−0.282 (−0.692, 0.128)	0.169	−0.303 (−0.723, 0.117)	0.149
MMP-12	−0.147 (−0.534, 0.241)	0.442	−0.098 (−0.49, 0.294)	0.611	−0.131 (−0.532, 0.271)	0.509
TIMP-1	−0.157 (−0.564, 0.251)	0.435	−0.221 (−0.626, 0.183)	0.269	−0.188 (−0.608, 0.231)	0.363
TIMP-2	−0.108 (−0.52, 0.303)	0.592	−0.133 (−0.545, 0.28)	0.513	−0.068 (−0.495, 0.359)	0.744

ROI—region of interest; FL—frontal lobe; PL-TL—parietotemporal lobe; POSC—posterior cingulate gyrus. ^1^ A *p*-value < 0.05 was considered statistically significant.

## Data Availability

Anonymized data will be shared at the reasonable request of qualified investigators.

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
