# Peer review of "Relationship between Cerebrospinal Fluid Matrix Metalloproteinases Levels and Brain Amyloid Deposition in Mild Cognitive Impairment"

_biomolecules, 2021, doi:10.3390/biom11101496_

Round 1
Reviewer 1 Report
This paper supports the finding that MMP-2 serves a protective function by preventing Aβ accumulation in the brain. The novel finding in this paper was that there is an association between MMP-7 in CSF and white matter lesions in patients with MCI.
Line 70: The sentence is incomplete. The supernatant was stored at -80°C until…
Lines 215-216: From the data presented in this paper, it is a stretch to say that MMP-7 is involved in blood-brain barrier disruption. Perhaps you could use MRI to examine blood brain barrier integrity in these patients.
General comments:
- Please discuss how the levels of PiB and FDG uptake in your study compare with those in other studies of patients with MCI. Also, how do the Fazekas scale in this study compare to other studies?
- Other studies (For example, Taniguchi et al., 2017; PMID: 28871443) have found that MMP-7 degrades Aβ1-42. Why do you think MMP-7 is not negatively correlated with Global PiB uptake in your study?
- I think it would be valuable to have supplementary figures/tables to compare the results from patients with different Apolipoprotein E Phenotypes as well as patients with an without diabetes, hypertension and hypercholesterolemia.
- One would need to be cautious in using MMPs as therapeutic targets since there is a fine balance between their protective functions (degrading amyloid beta) and detrimental functions (disrupting the blood brain barrier). This should be emphasized in the Discussion.
Author Response
We would like to take this opportunity to express our sincere thanks to the reviewers who identified areas of the manuscript that required corrections or modification. Based on the instructions provided in the decision letter and the reviewers’ comments, we have thoroughly revised the manuscript by modifying the relevant sections. Please find our point-by-point responses to the comments raised by the reviewers below. We hope that our revisions, along with our responses, have fully addressed the reviewers’ concerns and that our revised manuscript is now suitable for publication in Biomolecules.
REVIEWER COMMENTS:
Reviewer 1
We thank Reviewer #1 for the critical comments and useful suggestions, which have substantially helped us in improving our manuscript. As indicated in the responses below, we have considered each comment and revised the manuscript accordingly. We hope that our responses and revisions have fully addressed your concerns.
- Reviewer comment
Please discuss how the levels of PiB and FDG uptake in your study compare with those in other studies of patients with MCI. Also, how do the Fazekas scale in this study compare to other studies?
Response
We agree with the reviewer’s comment. We have revised the relevant sentences in the Introduction and Discussion sections, as follows:
P 2, line 53–58
Structural and functional imaging techniques are useful for identifying individuals at increased risk of developing AD. In particular, PiB-PET can detect brain amyloid deposition and predict conversion from MCI to AD [21,22]. WMLs on MRI or CT are frequently observed in patients with AD and influence cognitive function or cerebral perfusion [23,24]. BBB disruption and chronic hypoperfusion are the main causes of WMLs [25,26].
P 7, line 231–237
Of 33 subjects, 17 (51.5%) had higher PiB uptake and 20 (60.6%) had WMLs. The frequency of higher amyloid deposition is reported to be 54.6% in individuals with MCI aged 75–80 years [33] and that of WMLs to be 5%–87% in individuals aged 65 years and older, depending on the clinical characteristics and the method used to assess WMLs [34]. Moreover, two prior studies reported that WMLs were observed in approximately 70–80% of patients with AD [35,36]. Therefore, the frequency of individuals with higher PiB uptake or WMLs in the present study is consistent with findings from previous research.
- Reviewer comment
Other studies (For example, Taniguchi et al., 2017; PMID: 28871443) have found that MMP-7 degrades Aβ1-42. Why do you think MMP-7 is not negatively correlated with Global PiB uptake in your study?
Response
We agree with the reviewer’s comment. Taniguchi et al. reported that MMP-7 can degrade Aβ1-42. However, our results showed no correlation between MMP-7 and brain amyloid deposition. Therefore, we have added the following sentences to the Discussion section:
P 8, line 266–270
Although a previous in vitro study reported that MMP-7 may degrade Aβ1-42 to result in prevention of Aβ aggregation [48], our results showed no correlation between MMP-7 and brain amyloid deposition. One possible explanation for this discrepancy is that PiB-PET mainly detects insoluble fibrillar Aβ deposits [49].
- Reviewer comment
I think it would be valuable to have supplementary figures/tables to compare the results from patients with different Apolipoprotein E Phenotypes as well as patients with and without diabetes, hypertension and hypercholesterolemia.
Response
We agree with the reviewer’s comment. It is valuable to compare MMP levels between individuals with and without APOE4, as well as between individuals with and without vascular risk factors including diabetes, hypertension, and hypercholesterolemia. Therefore, we have added the Supplementary Tables and revised the relevant sentences in the Results section as follows:
P 5, line 180–183
There were no significant differences in CSF levels of MMPs and TIMPs between individuals with and without APOE4, or between individuals with and without vascular risk factors (Supplementary Tables 1 and 2).
Supplementary Table 1. Comparison of CSF MMP and TIMP levels between APOE4-negative and APOE4-positive groups
|
APOE4-negative (n = 22) |
APOE4-positive (n = 11) |
p |
MMP-2 |
7795.1 ± 2404.5 |
7997.1 ± 3525.7 |
0.955 |
MMP-7 |
54.2 ± 25.3 |
65.9 ± 34.4 |
0.440 |
MMP-12 |
34.7 ± 41.3 |
59.7 ± 107.2 |
0.380 |
TIMP-1 |
26872.9 ± 8218.7 |
29235.7 ± 8665 |
0.355 |
TIMP-2 |
23585.7 ± 6007.7 |
25415 ± 6498.2 |
0.778 |
MMP, matrix metalloproteinase; TIMP, tissue inhibitor of metalloproteinase. 1 A p-value < 0.05 was considered statistically significant.
Supplementary Table 2. Comparison of CSF MMP and TIMP levels between individuals with and without vascular risk factors.
|
Vascular risk factor (+) (n=27) |
Vascular risk factor (-) (n=6) |
p |
MMP-2 |
8026.5 ± 2983.7 |
7124 ± 1452.8 |
0.733 |
MMP-7 |
56.3 ± 28.3 |
66.2 ± 31.6 |
0.508 |
MMP-12 |
47.4 ± 75.7 |
23.4 ± 26.8 |
0.545 |
TIMP-1 |
27971.1 ± 8881.2 |
26262.7 ± 5388.1 |
0.600 |
TIMP-2 |
24209.8 ± 6582.2 |
24131.3 ± 3956.7 |
0.982 |
MMP, matrix metalloproteinase; TIMP, tissue inhibitor of metalloproteinase. 1 A p-value < 0.05 was considered statistically significant.
- Reviewer comment
One would need to be cautious in using MMPs as therapeutic targets since there is a fine balance between their protective functions (degrading amyloid beta) and detrimental functions (disrupting the blood brain barrier). This should be emphasized in the Discussion.
Response
We agree with the reviewer’s comment. Caution is required when using MMPs as therapeutic targets because they have both protective and detrimental functions. Therefore, we have revised the relevant sentences in the Discussion section as follows:
P 8, line 276–279
However, activation of MMPs appears to have both a beneficial role in Aβ catabolism and a detrimental role in BBB integration [53]. Therefore, we emphasize that caution must be taken when using MMPs as a therapeutic target due to their complex role in the pathology of AD.
- Reviewer comment
Line 70: The sentence is incomplete. The supernatant was stored at -80°C until…
Response
We have corrected the incomplete sentence in the Materials and Methods section as follows:
P 2, line 77–78
CSF samples were collected after centrifugation at 1500 × g for 10 min. The supernatant was frozen and stored in 1 mL aliquots at −80 °C until use.
- Reviewer comment
Lines 215-216: From the data presented in this paper, it is a stretch to say that MMP-7 is involved in blood-brain barrier disruption. Perhaps you could use MRI to examine blood brain barrier integrity in these patients.
Response
We apologize that we incorrectly described the methods used to evaluate white matter lesions. White matter lesions were evaluated in 30 individuals using MRI and in 3 individuals using CT. However, there were no changes in the conclusion.
To our knowledge, the integrity of the blood–brain barrier is commonly measured by contrast-enhanced MRI. In the present study, we did not perform contrast-enhanced MRI. Therefore, we have revised the relevant sentences in Methods and Results sections as follows:
P 2, line 67–68
Evaluation of cognitive function, brain MRI or CT, PiB- and FDG-PET, and CSF measurements of MMPs and TIMPs were performed in all individuals.
P 2, line 88–89
2.3. White matter lesion assessment
WMLs were evaluated by MRI for 30 individuals and by CT for 3 individuals.
P 2, line 93 to P 3, 96
Non-enhanced brain CT was performed with 3 mm continuous slices using Biograph 40 (Siemens, Erlangen, Germany). The severity of WMLs was assessed on T2-weighted images or CT scans using the Fazekas scale [28], according to previous studies [15,24].
P 5, line 175–177
Seventeen (51.5%) individuals were included in the higher PiB subgroup based on a PiB-PET SUVR cutoff of 1.4, while 20 (60.6%) individuals showed WMLs on MRI or CT.

Reviewer 2 Report
Dear authors, This is a timely paper on a study that addresses a very important issue in dementia science. It is vital to understand the continuum from MCI-related alterations to Alzheimer-type dementia. With the following additional analyses, along with minor improvements to the text and references, this will be an important study to help understand the MCI-AD connection.
- A clear statement of the hypothesis being tested needs to be included in the abstract and/or introduction, and end of abstract should focus on the MCI condition assessed (or extend to the MCI to AD continuum).
- For reader clarity, the correlation coefficient and corresponding full p value should be included in the legend for each figure component that assesses correlational analysis.
- For appropriate description of measures conducted, change the listing of “brain function” to the more appropriate description of evaluating glucose metabolism for cortical activity.
- For key measures being evaluated, it would be most appropriate and conventional to show example images (one with a lowest assessed measure found and one with a high assessed measure; or low, middle, and high level images) for the MRI imaging described and also for the PiB-PET and FDG-PET images utilized for quantitative evaluations.
- To improve the Discussion, give ref examples of directions towards treatments to slow the MCI to AD continuum. For example,
Marasco 2020. Current and evolving treatment strategies for the Alzheimer disease continuum. Am J Manag Care. 26:S167-S176. 10.37765/ajmc.2020.88481
Hwang et al. 2019 The role of lysosomes in a broad disease-modifying approach evaluated across transgenic mouse models of Alzheimer's disease and Parkinson's disease and models of mild cognitive impairment. International J Mol Sci 20:4432.
Kitajima et al.: Clinical impact of 11C-Pittsburgh compound-B positron emission tomography in addition to magnetic resonance imaging and single-photon emission computed tomography on diagnosis of mild cognitive impairment to Alzheimer's disease, Medicine: January 22, 2021
In particular, add to previous discussions regarding 11PiB-PET alone being useful for selecting patients who will progress from MCI to AD in the future, thus to govern treatment initiation.
- Finally, missing analyses should be shown or mentioned: control subjects (or any comparative literature of such subjects of the measures assessed), two-way ANOVA of the ApoE4 11 subjects vs the others (or most comparable group based on age/sex/health issues), and two-way ANOVA of female vs. males (using most comparable group based on age/health issues).
Author Response
We would like to take this opportunity to express our sincere thanks to the reviewers who identified areas of the manuscript that required corrections or modification. Based on the instructions provided in the decision letter and the reviewers’ comments, we have thoroughly revised the manuscript by modifying the relevant sections. Please find our point-by-point responses to the comments raised by the reviewers below. We hope that our revisions, along with our responses, have fully addressed the reviewers’ concerns and that our revised manuscript is now suitable for publication in Biomolecules.
Reviewer #2
We thank Reviewer #2 for the critical comments and useful suggestions, which have substantially helped us to improve our manuscript. As indicated in our responses below, we have considered each comment and suggestion and have revised the manuscript accordingly. We hope that our responses and revisions have addressed the reviewer’s concerns and that the revised manuscript is now suitable for publication.
- Reviewer comment
A clear statement of the hypothesis being tested needs to be included in the abstract and/or introduction, and end of abstract should focus on the MCI condition assessed (or extend to the MCI to AD continuum).
Response
We agree with the reviewer’s comment. Therefore, we have revised the relevant sentences in the Abstract and Introduction as follows:
P 1, line 11–14
This study aimed to explore whether cerebrospinal fluid (CSF) levels of matrix metalloproteinases (MMPs) and their inhibitors (TIMPs) were associated with brain amyloid deposition, cortical glucose metabolism, and white matter lesions (WMLs) in individuals with amnestic mild cognitive impairment (MCI).
P 1, line 24–26
These findings provide valuable insights into the role of MMPs in amyloid β catabolism and blood-brain barrier integration at the MCI stage.
P 2, line 58–60
We hypothesize that CSF levels of MMPs and TIMPs may be associated with brain amyloid deposition and BBB disruption, resulting in white matter degeneration at the MCI stage.
- Reviewer comment
For reader clarity, the correlation coefficient and corresponding full p value should be included in the legend for each figure component that assesses correlational analysis. Response
We agree with the reviewer’s comment. Accordingly, we have added the correlation coefficient and p value to the legend of each figure, as follows:
P 6, line 21–222
Figure 4. Relationship between CSF levels of MMPs and PiB uptake and Fazekas scale score. (A) MMP-2 CSF levels are negatively correlated with global PiB uptake (b = −0.414; 95% confidence interval, −0.796 to −0.032, p = 0.035). (B) MMP-7 CSF levels are positively correlated with Fazekas scale score (b = 0.419; 95% CI, 0.036 to 0.802, p = 0.033).
P 7, line 224–227
Figure 5. Relationship between CSF levels of MMP-2 and PiB uptake in the brain ROI. CSF levels of MMP-2 are negatively correlated with PiB uptake in the (A) posterior cingulate gyrus (b = −0.399; 95% CI, −0.782 to −0.017, p = 0.041) and (B) parietotemporal lobe (b = −0.476; 95% CI, −0.856 to −0.096, p = 0.016). POSC, posterior cingulate gyrus; PL-TL, parietotemporal lobe.
Reviewer comment
For appropriate description of measures conducted, change the listing of “brain function” to the more appropriate description of evaluating glucose metabolism for cortical activity. Response
We agree with the reviewer’s comment. Accordingly, we have revised the relevant sentences in the Abstract, Introduction, and Discussion as follows:
P 1, line 11–14
This study aimed to explore whether cerebrospinal fluid (CSF) levels of matrix metalloproteinases (MMPs) and their inhibitors (TIMPs) were associated with brain amyloid deposition, cortical glucose metabolism, and white matter lesions (WMLs) in individuals with amnestic mild cognitive impairment (MCI).
P 2, line 60–62
Therefore, this study aimed to examine the potential associations among CSF levels of MMPs and TIMPs, brain amyloid deposition, cortical glucose metabolism, and WMLs in individuals with amnestic MCI.
P 7, line 228–231
To the best of our knowledge, this study is the first to investigate whether CSF levels of MMPs and TIMPs are associated with brain amyloid deposition, glucose
metabolism, and WML severity in individuals with amnestic MCI.
- Reviewer comment
For key measures being evaluated, it would be most appropriate and conventional to show example images (one with a lowest assessed measure found and one with a high assessed measure; or low, middle, and high-level images) for the MRI imaging described and also for the PiB-PET and FDG-PET images utilized for quantitative evaluations.
Response
We agree with the reviewer’s comment. Therefore, we have added the following example images of the Fazekas scale scores, lower and higher PiB uptake on PiB-PET, and lower and higher FDG–PET uptake on FDG–PET:
P 3, line 108–112
Figure 1. Example of a Fazekas scale score of 0, 1, and 2 periventricular hyperintensity (top) and deep white matter hyperintensity (bottom) on T2-weighted images.
P 3, line 102–103
In this study, the Fazekas scale was determined as the sum of the periventricular and deep WMLs scores (Fig. 1).
P 4, line 150–153
Figure 2. Examples images of an individual with lower PiB uptake (A: cortical SUVR value = 0.804) and with higher PiB uptake (B: cortical SUVR value = 2.584).
P 4, line 154–158
Figure 3. Examples images of an individual with lower FDG uptake (A: cortical SUVR value = 0.709) and with higher FDG uptake (B: cortical SUVR value = 0.874).
P 4, line 145–147
The global standardized uptake value ratio of FDG-PET and PiB-PET was represented as a single mean value for all regions combined (Fig. 2 and Fig. 3).
- Reviewer comment
To improve the Discussion, give ref examples of directions towards treatments to slow the MCI to AD continuum.
Response
We agree with the reviewer’s comment. Therefore, we have added the relevant sentences to the Discussion section as follows:
P 8, line 271–279
Effective disease–modifying drugs are urgently needed to prevent the onset or slow the progression of AD. Potential targets for novel AD drugs include amyloid, tau, inflammation, metabolism, and neuroprotection [50]. Moreover, upregulation of lysosomal hydrolase and cathepsin B has been identified as a potential disease-modifying therapy in transgenic mouse models of AD [51,52]. Our results suggest that MMPs are a potential therapeutic target for slowing the progression of MCI to AD. However, activation of MMPs appears to have both a beneficial role in Aβ catabolism and a detrimental role in BBB integration [53]. Therefore, we emphasize that caution must be taken when using MMPs as a therapeutic target due to their complex role in the pathology of AD.
Round 2
Reviewer 1 Report
Thank-you for addressing my previous comments.
Author Response
Thank you for valuable comments.
Reviewer 2 Report
Please address Figure 4B: The B graph indicates p=0.081 therefore not significant and the text regarding positive correlation would need to be removed. However, the 4B legend indicates a different p value.
Please correct Figure 5: the p values within the graphs do not match those in the legend. If two different statistical analyses are being used, explain which is which.
Author Response
We would like to take this opportunity to express our sincere thanks to the reviewers who identified areas of the manuscript that required corrections or modification. Based on the instructions provided in the decision letter and the reviewers’ comments, we have thoroughly revised the manuscript by modifying the relevant sections. Please find our point-by-point responses to the comments raised by the reviewers below. We hope that our revisions, along with our responses, have fully addressed the reviewers’ concerns and that our revised manuscript is now suitable for publication in Biomolecules.
Reviewer #2
We thank Reviewer #2 for the critical comments and useful suggestions, which have substantially helped us to improve our manuscript. As indicated in our responses below, we have considered each comment and suggestion and have revised the manuscript accordingly. We hope that our responses and revisions have addressed the reviewer’s concerns and that the revised manuscript is now suitable for publication.
- Reviewer comment
Please address Figure 4B: The B graph indicates p=0.081 therefore not significant and the text regarding positive correlation would need to be removed. However, the 4B legend indicates a different p value.
Please correct Figure 5: the p values within the graphs do not match those in the legend. If two different statistical analyses are being used, explain which is which.
Response
We showed the results of Spearman's rank correlation coefficient in the Figure 4 and Figure 5 for the explanation of figures. However, we performed multiple regression model to investigate the relationship among the CSF levels of MMPs and TIMPs, the PiB and FDG uptake values, and Fazekas scale scores after adjusting for covariates, such as age, sex, education level, frequency of each vascular risk factor, and APOE4 status. Therefore, we remove the p values in these figures.